# The Results After One Year of an Experimental Protocol Aimed at Reducing Paratuberculosis in an Intensive Dairy Herd

**DOI:** 10.3390/ani15182695

**Published:** 2025-09-15

**Authors:** Anita Filippi, Giordano Ventura, Antonella Lamontanara, Luigi Orrù, Fabio Ostanello, Riccardo Frontoni, Laura Mazzera, Edoardo Tuccia, Matteo Ricchi, Chiara Garbarino

**Affiliations:** 1National Reference Center for Paratuberculosis, Experimental Zooprophylactic Institute of Lombardy and Emilia-Romagna (IZSLER), “Bruno Ubertini”, Diagnostic Section of Piacenza, Via Strada Della Faggiola 1, 29027 Podenzano, PC, Italy; matteo.ricchi@izsler.it (M.R.); chiaraanna.garbarino@izsler.it (C.G.); 2Experimental Zooprophylactic Institute of Lombardy and Emilia-Romagna (IZSLER), “Bruno Ubertini”, Diagnostic Section of Cremona, Via Cardinale Guglielmo Massaia 7, 26100 Cremona, CR, Italy; giordano.ventura@izsler.it; 3Council for Agricultural Research and Economics (CREA)—Research Centre for Genomics & Bioinformatics, Via S. Protaso 302, 29017 San Protaso, PC, Italy; antonella.lamontanara@crea.gov.it (A.L.); luigi.orru@crea.gov.it (L.O.); 4Department of Veterinary Medical Sciences, University of Bologna, 40064 Ozzano dell’Emilia, BO, Italy; fabio.ostanello@unibo.it (F.O.); riccardofrontoni@outlook.it (R.F.); 5Risk Analysis and Genomic Epidemiology Unit, Experimental Zooprophylactic Institute of Lombardy and Emilia Romagna (IZSLER), Via dei Mercati 13/A, 43126 Parma, PR, Italy; laura.mazzera@izsler.it; 6Bovine Practitioner, 26900 Lodi, LO, Italy; edoardo.tuccia@gmail.com

**Keywords:** MAP, paratuberculosis, cattle, digital PCR, quantification

## Abstract

Paratuberculosis, caused by *Mycobacterium avium* subsp. *paratuberculosis* (MAP), is a chronic incurable enteritis of ruminants, globally responsible for economic losses and compromising animal health and welfare. Infected animals can spread the bacteria for years without showing symptoms, contaminating the environment and potentially infecting other animals. In this study, we aimed to reduce the number of infected animals in a paratuberculosis-infected dairy farm by combining laboratory testing and changes in farm management. All adult cows were tested to identify infected animals and MAP amount released by each infected cow was quantified, helping the farmers to decide which animals to remove first. At the same time, management improvements were made. The results showed a reduction in the prevalence of paratuberculosis in a relatively short time on the farm. This approach could help to improve animal health.

## 1. Introduction

Paratuberculosis is a chronic, incurable enteritis of ruminants sustained by *Mycobacterium avium* subsp. *paratuberculosis* (MAP), which is globally responsible for huge economic losses in the livestock sector, as well as compromising animal health and welfare.

The global cost of paratuberculosis is estimated to be as high as 4 billion USD per year [1]; this has had a particularly strong impact on the dairy sector. Although to date no final consensus has been reached by the scientific community with respect to MAP’s zoonotic potential, its involvement in the etiopathogenesis of some human diseases (Crohn’s disease, type I diabetes, multiple sclerosis, Hashimoto’s thyroiditis) has been speculated [2,3]. In this regard, there is also concern regarding MAP’s ability to contaminate products in the milk and meat supply chain [4,5]. There are several factors that make managing the disease on the farm challenging: infected animals remain indistinguishable from uninfected ones for a long time [6]; MAP is excreted in feces and characterized by high resistance in the environment, factors that make the enforcement of biosecurity standards crucial [7]; diagnostic tests have strong limitations, notably, low sensitivity at many stages of disease; at the moment no effective treatments are available [8]; vaccines exist, but they interfere with the intradermal reaction applied in bovine tuberculosis eradication programs, so, in countries were tuberculosis control plans are active, such as Italy, these vaccines cannot be used in cattle [6].

Infected animals spread MAP in the environment mainly through feces, in both clinical and subclinical stages of infection. The ingestion of food, milk, and water contaminated with infected feces is the main factor in the transmission of the disease [9,10]. As an obligate parasite, MAP does not multiply outside of host species; however, it is characterized by an extremely high environmental resistance [11].

MAP has a variable incubation period of up to 2–7 years [6], a period in which animals may not show symptoms despite being infected; only 10–15% of animals develop clinical signs of the disease after 24 months of age [10,12] and the evolution of the disease involves different stages: latent, subclinical, clinical, and advanced. While the amount of MAP excreted is very low in the early stages of infection, and consequently the risk of infecting other animals is low, as the disease progresses, the amount of MAP excreted increases, thus increasing the risk of exposure for healthy animals [13,14].

Therefore, both the prevalence of infected animals on the farm and the MAP load excreted by them into the environment are crucial for the level of environmental biocontamination. In particular, the main contributors to the contamination of the environment and the food chain are the so-called “Super-Shedders”, which excrete up to 10^4^ CFU/g of MAP in feces [15]. Under conditions of high contamination of the housing environment, the presence of passive eliminator animals, i.e., cattle in which the infection has not been established but excrete low MAP loads within their feces (passing-through), is also possible [16,17,18]. In this regard, the quantification of MAP eliminated through feces allows the prioritization of on-farm interventions (removal of Super Shedders), managing infected animals in relation to their potential for environmental biocontamination and avoiding the reformation of possible Passive Shedders.

Although some reports have shown that paratuberculosis can be successfully eradicated in goats and cattle [19,20], the disease remains very difficult to control. It requires improvements in biosecurity within herds, the control of purchased animals, and the application of test-and-cull strategies [19].

The aim of this study was to reduce the prevalence of paratuberculosis in a single intensive dairy herd by combining diagnostic laboratory outputs and management procedures. In more detail, from the diagnostic point of view, it was possible to apply fecal PCR and serological ELISA in parallel to all cows over 24 months present in the herd. Moreover, to prioritize which animals should be removed first, a dPCR was employed in the field to quantify the MAP load in each fecal-positive cow. This expensive protocol was partially funded by public resources. Conversely, the owner allocated significant financial resources to improving all management procedures, including structural changes to the farm, and covered the cost of eliminating infected animals.

## 2. Materials and Methods

### 2.1. Study Farm

A herd located in the Lombardy region, Northern Italy, one of the areas with the highest density of dairy farms in the country (around 0.21 farms per km^2^ and 47.39 heads per km^2^, the highest in Italy (data provided by the Zootechnical Registry established by the Italian Ministry of Health at the National Service Centre for Animal Identification and Registration Database, Experimental Zooprophylactic Institute of Abruzzo e Molise; https://www.vetinfo.it/j6_statistiche/#/, last access 27 June 2025), was selected. The herd had a high prevalence history of paratuberculosis: at a control carried out on November 2022, the farm showed a 17% apparent serological prevalence (evaluated by testing all adult animals ≥ 24 months). After this, the majority of those serological positive animals were removed from the herd, and no further diagnostic protocols were applied. The herd was an intensive Friesian dairy herd, with free stalling, about 1000 lactating cows, and mainly internal replacement. The milk produced was destined for cheese-making.

At the beginning of the study, the facilities were not particularly modern, environmental hygiene was poor, and animal density was high according to Classy-Farm checklist parameters. Classy-Farm is an Italian Ministry of Health system that monitors farms. It integrates data on animal welfare, antibiotic use, biosecurity, and animal health to classify them by risk (https://www.classyfarm.it/index.php/news-it/nuove-checklist-e-manuali-di-autocontrollo-per-la-valutazione-del-benessere-animale-e-della-biosicurezza-nei-bovini, last access 27 June 2025). Figure 1 shows the representation of the farm layout at the first inspection.

### 2.2. Samples and Data Collection

Within the scope of this research, three visits were carried out at the farm over approximately 24 months; during the first visit, an evaluation of the risk of transmission of paratuberculosis, according to the “Manual for the control of Paratuberculosis in cattle and dairy buffalo herds” (https://www.izsler.it/cdrparatubercolosi/wp-content/uploads/sites/11/2023/06/NEL-04_Allegato-1-A_MANUALE-bovine_bufale-latte_Rev_2.pdf) was carried out.

A second evaluation of the risk of the transmission of paratuberculosis was performed during the third visit.

During the first two visits, the following samples were taken: (1) feces from the rectal ampulla and blood for serum from the tail vein from every animal aged > or =24 months; (2) environmental samples by taking stool from the end points of the scrapers and areas of different housing environments; (3) bulk milk from the different milking groups. During the third inspection, blood samples from the tail vein were taken from all animals aged ≥ 24 months, and environmental stool and bulk milk samples were also taken. Due to budget constraints, fecal sampling during this third visit was limited to selected animals. Specifically, from the animals that tested positive for serology at this last sampling and from other animals that had been indicated as “to be under monitoring”, in relation to previous results, fecal samples were taken from the rectal ampulla and sent to the laboratory for PCR and cultural analyses. Animals that were defined as “to be under monitoring” were those that were potential “Passive Shedders”, i.e., animals that had previously tested positive or inconclusively in PCR tests, but negative in ELISA tests. The “To be under monitoring” group also included animals that had entered the over-24-month group at the date of the third inspection. The number of animals sampled varied depending on their age and whether they were present in the herd at the time of sampling. A total of 3104 blood, 2136 individual fecal, and 27 environmental samples were analyzed. Specifically, 1019 individual blood and feces samples were taken and analyzed in May 2023, as well as 970 blood and fecal samples in November 2023. Finally, 1115 blood samples were collected in June 2024 and, after two days, 147 fecal samples were collected.

At the same time, an environmental convenience sampling was carried out during the three visits. Eleven samples were collected in May 2023, seven samples in November 2023, and nine samples in June 2024. The areas sampled encompass scrapers and different paddocks (husbandry, heifers’ area, etc.). The aim of environmental sampling was to collect environmental MAP field isolates for comparative purposes by whole genome sequencing analysis (WGS). For bulk milk, six samples were taken in May 2023, six in November 2023, and one in June 2024.

### 2.3. Serology

Once they arrived at the laboratory, the samples were allowed to clot at room temperature before being centrifuged and analyzed within two days. For the last sampling, considering the high quantity of samples, sera were processed using an automatic robot that allowed the rapid processing of the sera samples. The ELISA test used was a commercial kit (ID Screen^®^ Paratuberculosis Indirect and Id Confirmation^®^ Paratuberculosis indirect, both ID VET, Montpellier, France). The assay is based on a screening test followed by a confirmation one. In more detail, after a pre-adsorption step with *Mycobacterium phlei*, a screening ELISA was used to discriminate positive and inconclusive samples from negative ones (S/P ratio ≤ 0.6); inconclusive (0.6 < S/P ratio < 0.7) and positive (S/P ≥ 0.7) samples from the screening test were then submitted to confirmatory ELISA test, in order to verify the specificity of the reaction. Samples with an S/P ratio ≥ 0.6 in the confirmatory test were considered positive. The manufacturer reports a specificity over 99% and a general sensitivity around 57%.

### 2.4. IS900 qPCR

The qPCR used in the project, targeting the *IS900* insertion sequence, has been validated according to the guidelines provided by WOAH [21]. In detail, DNA was extracted from 3 ± 0.5 g of feces and qPCR was performed on eluted DNA, targeting *IS900* sequence, while analyzing samples in duplicate [21]. Results were expressed by the cycle of quantification (Cq) values; a sample was considered positive if both replicates resulted in ≤36 Cq, negative if both replicates were >40, and inconclusive if one replicate resulted in ≤36 and one >36 or if both replicates were between 36 and 40. Only the analysis of inconclusive samples was repeated and, if the same result was confirmed, the final result was inconclusive. The performance of the test is reported in Russo et al. [21].

### 2.5. Digital PCR (dPCR)

For the direct quantification of the number of MAP present in feces, the previously obtained DNA was analyzed by digital PCR according to Russo et al. [22], using the QIAcuity One system (Qiagen) with 24-sample plates with 8500 and 26,000 partitions. Digital PCR provides the possibility of directly quantifying the number of MAP cells from the fecal sample, offering several advantages over the culture method: reduction of analysis time from months to days, overcoming the problem of cultural contamination, and offering the possibility of quantifying difficult-to-cultivate field isolates. Moreover, the quantification of excreted MAP does not require specific standards and it is considered in general more robust than qPCR [23]. In this work, animals that eliminated a high concentration of MAP, defined as ≥10^7^ MAP genome copies/g feces, were considered Super Shedders/eliminators; animals that eliminated from 10^5^ to 10^7^ MAP genome copies/g feces were considered High Shedders/eliminators. Because of the limit of quantification fixed for the method (around 10^5^ MAP cell/g of feces), it was not possible to classify animals shedding lesser amounts of MAP with the appropriate precision (see Russo et al., 2023 [22] for more details). The relation between colony forming units and genome copies was investigated elsewhere [23].

### 2.6. Bacteriology

To further confirm the presence and viability of MAP circulating in the herd and in order to obtain field isolates of MAP for epidemiological investigations, samples positive to qPCR were submitted to culture. The bacteriological method was performed on 3 ± 0.5 g of feces according to the double incubation method reported in the WOAH Manual for terrestrial animal Chapter 3.1.17 [9]. Final suspensions were incubated in homemade Herrold’s egg yolk medium (HEYM), containing 2 mg of mycobactin J/L, supplemented with Chloramphenicol (30 mg/L) (HEYM/CAF) or with Nalidixic acid (50 mg/L), Vancomycin (50 mg/l), and Sodium Pyruvate (4 g/L) (HEY-MPANV). Suspected positive colonies were confirmed as previously reported [24]. The limit of detection of the method was assessed around 10^3^ CFU/g of feces. Results are reported considering the number of colony-forming units (CFU) and are expressed as follows: 1+, Low Shedders (1–9 CFU/slant); 2+, Moderate Shedders (10–49 CFU/slant); 3+, High Shedders (over 50 CFU/slant) [21]. After the first visit, 52 different field isolates, 45 from individual cows and 7 from environmental samples, were obtained; after the second and the third visits, 18 isolates and 10 isolates, respectively, were obtained from individual cows, while 4 and 3 environmental MAP isolates were obtained after the second and the third visit, respectively.

### 2.7. Whole Genome Sequencing

Because of the difficulties in obtaining pure cultures in the appropriate amount of MAP field isolates, only 31 MAP isolates were submitted to whole genome sequencing and analyzed according to Bolzoni et al. [25]. More specifically, 22 isolates were obtained from individual cows and 1 from environmental sampling after the first visit, and 7 were obtained from individual cows plus 1 from environmental sampling after the third visit.

Data were processed according to the following bioinformatic pipeline: Illumina raw reads were trimmed using Trimmomatic v0.39 software [26] to remove low-quality reads using a sliding window quality cut-off of Q20 in 20 bp. Trimmed reads were corrected with Karect v1.0 [27]. To identify the SNPs, the corrected reads were first mapped against the MAP genome K10 using Bowtie2 v kc2.5.1 [28], generating the SAM files. Using Samtools v1.20, the SAM files were converted into BAM files and sorted. Using the mpileup command from all sorted BAM files, a text pileup output was generated. Variants were designated using VarScan v2.4.6 with the following parameters: min-coverage 15, min-reads 25, min var freq 0.7, min-avg-qual 20, and *p*-value thresh 0.01. Only polymorphic sites present in all genomes were considered. The SNPs inferred from the core genome were used to reconstruct a maximum likelihood (ML) phylogenetic tree using IQ-Tree v.2.2.2.3 [29]. The best-fit models of nucleotide substitution were identified using Model Finder as implemented in IQ-Tree based on the Bayesian information criterion (BIC). The IQ-TREE analyses were performed using the K2P+ASC substitution model. The consensus phylogenetic tree is constructed from 1000 bootstrap trees. The phylogenetic tree was drawn using the ggtree v3.8.2 in R package [30] within the R version 4.3.0. Genetic relatedness among isolates was assessed by determining pairwise SNP differences using the program snp-dist. The distance matrix that resulted from snp-dist analysis was plotted as a heatmap using the pheatmap R package v1.0.13. The whole genome sequencing data collected for this study were deposited in the European Nucleotide Archive under Bioproject number PRJEB96154.

### 2.8. Statistical Analysis of Apparent Seroprevalences

The apparent seroprevalences of paratuberculosis and their 95% confidence intervals (CI_95%_) were calculated using the exact binomial test in the R Studio package (version 1.4.1106). Statistical differences were investigated by two-way chi-square test using MedCalc Software Ltd. Version 23.3.5 (Ostend, Belgium, https://www.medcalc.org/calc/chisquared-2way.php) (accessed on 19 August 2025).

## 3. Results

### 3.1. First Inspection of the Herd for the Evaluation of the Risk of Transmission of Paratuberculosis

With regard to the risk assessment during the first inspection (May 2023), several critical points were found, especially related to biosecurity: (1) direct and indirect contact between young and adult animals; (2) absence of dedicated equipment for the different categories of animals (forks, boots, etc.); (3) poor environmental hygiene, in particular poor cleaning of floors and walking areas not used for decubitus (corridors, passageways, exercise areas), along with poor cleaning of the housing areas and of troughs. In addition, we found some “risky” behaviors. In particular, the owner purchased 68 animals for replacement from Germany in January 2023 and 16 animals in May 2023 from a surrounding herd. The introduced animals came from herds with no certifications regarding paratuberculosis, but which had instead merely been given a generic statement that paratuberculosis was not present in the herd.

### 3.2. First Sampling

After the first sampling in May 2023, 111 animals tested positive in at least one test (feces, serum, or both). In particular, 26 animals tested positive in the serological examination but negative in the PCR test; 33 animals tested positive in the PCR test but negative in the serological examination; and, finally, 33 animals tested positive in both tests (Table 1a). One hundred and seventy-four samples were PCR inconclusive. Of these, 19 tested positive for ELISA and the rest tested negative. In this paper, we have considered PCR-positive/-inconclusive and serology-negative animals to be possible Passive Shedders. Interestingly, all of these samples showed Cq values > 31. To determine whether these animals could be classified as Passive Shedders or positive in the early stages of the disease, we waited for the results of further sampling. All 66 PCR-positive individual samples in this first visit were submitted to dPCR. The animals were then classified as Super Shedders or High Shedders according to the results obtained (Table 2a). Notably, all but one Super Shedders animals were ELISA positive. After the first sampling, recommendations were provided to the farmer to prioritize the removing of all Super Shedders (six animals), followed by High Shedders (22 animals) if possible, who also tested positive for the ELISA test. Furthermore, recommendations were made to ensure that cows testing negative for serology, with weak positives (Cq over 31), or inconclusive by fecal PCR were not removed from the herd, being possible Passive Shedders. Regarding environmental samples, 7 out of 11 samples tested positive. Among the six bulk tank milk samples analyzed, one was inconclusive and the others tested negative.

### 3.3. Second Sampling

After checking the animals’ records, we found that all Super Shedders and 5 High Shedders, as well as 219 other animals, had been removed from the herd after the first visit.

Overall, the second sampling (November 2023) revealed that 58 animals tested positive for at least one of the tests (feces, serum, or both). Fourteen animals tested positive in both assays, twenty-three tested positive in serology but negative in PCR, eight tested positive in serology but inconclusive in qPCR, and thirteen tested positive in PCR but negative in serology (Table 1b). Furthermore, a total of 37 samples were inconclusive for PCR, including the 8 that were also positive for ELISA.

Among the cattle that tested positive for at least one test after the first visit, 31 were still present in the herd. Focusing on the cows that were positive for serology on the first visit (15 animals), three samples tested positive both by fecal PCR and ELISA assay. Two samples were serologically positive but PCR results were inconclusive. Four samples tested negative for both assays. Five samples tested positive for serology but negative for PCR. One sample tested negative for serology but inconclusive for PCR.

Among the cows that were positive by fecal PCR after the first visit (eighteen animals, excluding the samples judged inconclusive after the first visit), three animals were positive for both assays after this second visit, two animals were positive for serology but negatives for fecal PCR, two were negative for serology and positive for PCR, one was negative for serology and inconclusive for PCR, and the last ten were negative for both assays. The load of MAP of these five PCR positive cows was quantified by dPCR (Table 2 and Figure 2). Among these samples, only cow n. 706, considering the dPCR and serology results after the two different visits, showed clear evidence of any progression of the disease.

In detail, we observed an increment in the load of MAP (from around 2.09 × 10^4^ to 2.08 × 10^5^ cells/g of feces) and a change in the result of the serological assay from negative to positive. For cows no. 147 and 651, serological assays were negative after the first two visits, but the load remained approximately constant for cow no. 651 (load unquantifiable or very low) and decreased for cow no. 147 (from 4.20 10^5^ to 2.00 10^4^ MAP cells/g). Cows no. 81 and 810 showed the same result regarding MAP load as cow no. 651, but the ELISA test became positive after the second visit. Cow no. 1038 (see Figure 2) was not tested in the previous visit but was a High Shedder, as well as cows no. 1005, 365, and 488 (see Figure 2), which were instead negative after the first visit. Overall, no new Super Shedders but seven High Shedders were still found; one of these was cow n. 706. Notably, all these animals were also ELISA positive.

Twenty animals (positive or inconclusive for PCR but negative for ELISA after the first visit) were found to be positive/inconclusive after the second visit. These animals were not classified as Passive Shedders since they had tested positive/inconclusive to two consecutive PCRs.

Finally, 104 cows that tested positive or inconclusive for PCR and negative for serological assay after the first visit, but PCR negative and ELISA negative after the second visit, were considered as potential Passive Shedders. In this case, the Cq of these cows was over, or very close, to the cut-off of the PCR, suggesting a low level of MAP load.

Even after this visit, recommendations were provided to the farmer to prioritize the removal of these last animals, but the farmer decided to also send the remaining seropositive animals to the abattoir.

With regard to the environmental sampling, four out of seven samples tested positive. The six bulk milk samples analyzed were negative.

### 3.4. Second Inspection of the Herd for the Evaluation of the Risk of Transmission of Paratuberculosis

In June 2024, the third visit was carried out, preceded by a reassessment of the risk of the transmission of paratuberculosis. Compared to the first visit, some significant structural changes had been undertaken by the farmer in order to resolve some of the issues raised during the first visit. In particular, the replacement herd (calves and heifers, both to be fertilized and pregnant) was moved to another site, about 3 km away from the main area but belonging to the same property. After birth, the calves were transported to a new location, where they remained until their first insemination. They were then transported back to the main site. Male calves, on the other hand, remained on the farm and were then sold for meat production. Furthermore, during the various growth phases, in order to separate the more susceptible categories (young animals) from the adult cows, the animals were housed in homogeneous groups according to age. Other changes were also made regarding the management of the main farm: (1) division of the farrowing room into areas for seronegative and seropositive animals; (2) division of the sick pen in two separate areas—for housing animals with or without enteric forms—in order to avoid contact with potentially MAP-eliminating animals; (3) modification of colostrum management—only colostrum from animals that tested negative for both assays in the previous check was stored and provided to the animals.

### 3.5. Third Sampling

During the third visit in June 2024, out of the 1115 blood samples taken, 50 tested positive for the serological test (4.5%). Due to a lack of resources, it was not possible to perform another fecal sampling including all cows present in the herd, but 147 fecal samples, including those from seropositive cows, were analyzed by PCR. Out of 50 seropositive cows, 12 were also PCR positive, while only 1 seronegative cow was PCR positive after the third visit. Among the 12 cows that tested positive for both assays after this last visit, 2 animals were included in the testing for the first time, while 1 of the cows was negative to both assays in the previous visits, and the remainder of these 12 animals were positive or inconclusive for PCR and ELISA assays carried out on the previous visits (Figure 2). Among the remaining 38 seropositive but negative/inconclusive for PCR cows, 26 were positive either for ELISA or PCR in the previous visits.

The load of MAP was quantified by dPCR in all 13 fecal samples that tested positive for PCR (Table 2c). Four cows were labeled as High Shedders after this analysis, while the others did not return any results or their load was lower. Notably, all these last High Shedder cattle were ELISA negative.

In order to check for the presence of potential Passive Shedders after this last visit, only 94 cows remained in the herd from the previous visits. The ELISA test was never positive for these cows. On the first visit, 6 of them tested positive by PCR and 76 tested inconclusive, whereas, after the second visit, only 3 tested positive by PCR and 14 tested inconclusive. Overall, after subtracting those that tested positive or inconclusive even after the third visit, 82 of the last 94 cows were suspected to be Passive Shedders.

After the third visit, the median load of MAP obtained by dPCR was plotted by the ordinal classes of the cultural assay for the fecal samples positive to the cultural assay and having a number that was quantifiable by dPCR (see Appendix A, 55 samples considered). As expected, the MAP load obtained by dPCR was proportional to the number of colonies recovered by the cultural assay.

### 3.6. Paratuberculosis Apparent Seroprevalence During the Study

The health situation with regard to paratuberculosis, comparing the results of the serological examination, has improved since the first inspection (Figure 3). In particular, serological results referring to each of the three samplings are reported. At visit 1, the apparent serological prevalence was 7.6% (78 positive animals/1019 examined, CI_95%_: 6.1–9.4%); at visit 2, the apparent serological prevalence was reduced to 4.6% (45 positive animals/970 examined, CI_95%_: 3.4–6.1%). At visit 3, the apparent serological prevalence was 4.5% (50 positive animals/1115 examined, CI_95%_: 3.3–5.8%). The decrease in apparent serological prevalence among the three visits was statistically significant (chi-square: 9.9174; *p* = 0.0016). The two-by-two comparison found that the apparent prevalence at the first visit was statistically different from the other two, even considering the Bonferroni correction for multiple comparisons (*p ≤* 0.0167), but there was no statistically significant difference between visits 2 and 3 (*p* = 0.8657) (Figure 3).

### 3.7. Whole Genome Sequencing Analysis

MAP is a slow-growing pathogen characterized by low genetic diversity [31]. The standard approach for detecting transmission pathways relies on assigning isolates to putative transmission clusters. Applying an SNP threshold is a commonly used method to group isolates into the same cluster. In this regard, the relationships between MAP isolates were investigated using SNPs identified by aligning sequencing reads from the isolates to the MAP K-10 reference genome. In order to focus on the variability present in the core genome, only SNPs located within the genomic regions shared by all isolates were retained. Using stringent criteria, a total of 20 high-confidence SNPs were identified in the “core genome”. These SNPs were used to draw the heatmap in Figure 4 and build the phylogenetic tree in Figure 5. The pairwise SNP distance matrix revealed a close genetic relatedness among the MAP isolates recovered in this study, with all isolates exhibiting a pairwise SNP distance below five SNPs. Phylogenetic analysis confirmed the close similarity among the MAP field isolates.

## 4. Discussion

In the present study, with the aim of implementing a strict health management plan in a relatively short period of time (13 months, from May 2023 to June 2024), we have applied a diagnostic approach, consisting of testing all cattle over 24 months of age twice in parallel by both the serological ELISA test and fecal PCR. A third sampling was applied as well to all cattle over 24 months of age for serological purposes and, from a subgroup of cows that had been indicated as “to be under monitoring” on the basis of previous results, fecal individual samples were also analyzed by direct PCR. Fecal positive samples were also submitted to dPCR for the fecal quantification of MAP, with the final aim of prioritizing animals that should be removed or segregated. After each visit, reports with results of diagnostic tests, coupled with suggestions about how to manage positive cows and further information about all critical points relative to the risk of transmission of paratuberculosis, such as the implementation of environmental hygiene measures, were provided to the farmer. Such an extensive and expansive diagnostic protocol would not have been possible without the support of an internal project evaluating the performance of dPCR in the field, which covered some of the costs. In addition to this diagnostic program, the owner improved the management procedures and made the decision to cull or segregate positive cattle, investing a lot of money in these actions. Notably, according to the national guidelines for paratuberculosis, issued in 2013 by the Italian Ministry of Health [32,33], the farmer should have covered all costs for the diagnostic tests.

According to the guidelines, the health status of the herd is based on results of standardized serological testing schemes, which have to be repeated annually. The ELISA test, exclusively for economic reasons, is the test adopted by almost all plans aimed at controlling paratuberculosis worldwide [7]. The qualifications for cows and buffalos are reported in the National Data Bank (BDN). Notably, at the beginning of the study, the farm was classified as PT0 (i.e., not having been object of notification of paratuberculosis clinical cases); after the application of this protocol, the farmer could consider applying for PT1 qualification, which is possible to obtain with no clinical cases and an apparent serological prevalence ≤ 5% among all animals, including all the females aged over 36 months and all males and purchased animals aged over 24 months.

Paratuberculosis is estimated to affect 50% [4] of cattle herds worldwide, and the application of hygienic procedures associated with management practices to avoid the exposure of young animals to MAP, together with the earliest possible identification and timely culling of infected animals, are the main strategies to control paratuberculosis today [7,34,35]. In our case, one of the procedures implemented by the farmer was to relocate the young animal facility to a new area. This was not far from the original site, but far enough away to prevent direct or indirect accidental contact between the young and adult animals. Suggestions were also provided to minimize contact through fomites [36].

Another critical issue regarding the disease control is related to the low sensitivity of all diagnostic tests currently available at many stages of the disease [9,36], especially at early stages. In order to overcome the problem of the low sensitivity in diagnostic tests, strategies were proposed and modeled that employed these tests in parallel and in a longitudinal way, i.e., testing the same animal over a certain timeline [37,38,39].

The diagnosis of paratuberculosis has focused on qualitative information (infected/uninfected animal), using different methods: cultural assays, PCR to detect MAP DNA, or the ELISA serological technique for the detection of antibodies [9]. The cultural method is highly specific and is therefore designated by the World Organization for Animal Health (WOAH) as the gold standard for the diagnosis of paratuberculosis [21]; moreover, it was the only one of these methods that, when performed on solid medium, returned semi-quantitative results [16,19]. However, due to the slow growth of MAP, the cultivation process is complex and time-consuming [40]. Visible colonies only appear after four to eight weeks, and, in some cases, results can take months, providing very late feedback to be used for the rapid selection of animals for culling or segregation. In this study, a new diagnostic tool, digital PCR (dPCR), an innovative technology used for the absolute detection and quantification of nucleic acids without calibration standards [22], was used in the field for the first time. Through the rapid quantification of the number of MAP cells excreted in feces, the project assessed the risk of MAP spread by individual animals in order to prioritize on-farm interventions (removal of Super Shedders), managing animals in relation to their potential for environmental biocontamination, avoiding the reformation of possible Passive Shedders, and providing useful information for the application of effective and sustainable health management plans on dairy cattle farms, while reducing the contamination of the food chain.

In our study, we found that some animals became infected after previously testing negative. Some tested positive on just one assay and then tested positive or were confirmed positive after further sampling. Some tested negative on both serology and fecal PCR, while others tested positive on the first fecal PCR but negative over the years. Similar data were obtained in almost all longitudinal studies [38,40,41] and have helped to define the presence, in every single outbreak, of progressor animals (animals in which the disease progresses), animals able to contain the infection in some ways, and negative animals, those in which MAP was not able to establish an infection. Conversely, true positives were defined as animals that tested positive in both the ELISA and PCR assays, animals that continued to test positive after further sampling, or animals with a very strong positive result for at least one test (ELISA or fecal PCR).

Among negative animals, Passive Shedders are those in which MAP is detected in the feces, generally at low or very low levels, and tend to disappear when Super or High Shedder animals are removed from the herd [16,18]. The level of MAP excretion in feces differs widely between secretory infected animals. Indeed, the so-called “Super Shedder” animals can shed more than 10^4^ CFU/g of feces. According to a study, a “Super Shedder” is equivalent to more than 160 High Shedders, 2000 Moderate Shedders, and 20,000 Low Shedders [15]. Because of the presence of these animals within the herd, a high contamination of the housing environment is probable and this can explain the presence of “Passive Shedders”, i.e., animals in which the infection has not been established and which should never become seropositive [42], but which are able to eliminate low MAP loads in their feces [16,18,42]. In our case, almost 80 cows were suspected to be “Passive Shedders”, quite a high percentage of the total cattle present in the herd. Other authors reported how almost 50% of animals shedding low amounts of MAP can be classified as Passive Shedders, while the remaining 50% proved to be active MAP Shedders [42]. In this regard, both the number of infective animals present on the farm and the MAP load excreted by them in the environment are crucial for the level of environmental biocontamination [41,43]. Indeed, after the removal of “Super Shedders” and “High Shedders” following the first visit, the total number of inclusive fecal PCRs dropped from 174 to 37 (Table 1), even if, also after the second visit, 7 “High Shedders” cattle were still found within the herd, further confirming that many years of testing and management are necessary in order to control the disease. Some limitations of the above concept are related to the fact that we did not analyze potential “Passive Shedders” after slaughtering these animals, so we were unable to demonstrate the absence of MAP within the intestines of these animals, which is a fundamental requisite for defining “Passive Shedder” animals [16,18]. However, the strong reduction in the number of fecal PCR positive and inconclusive cows from the first to the second visits, coupled with the observation of a single clone circulating within the farm, suggest the presence of Passive Shedders.

In this regard, WGS analyses revealed a strong genetic relationship among the field MAP isolates recovered within the farm. A pairwise distance of below five SNPs indicates that the recovered MAP has a clonal origin, as has been suggested previously for MAP [25] and *Mycobacterium bovis* [44]. The recovery of closely related isolates in both cows and environmental samples appears to further confirm the presence of Passive Shedders [18,25]. The close relationship between the different isolates can be considered in two ways. Firstly, these data provide evidence of recent transmission events occurring among animals. Secondly, we were unable to detect different lineages among the isolates, even considering those coming from environmental samples, suggesting that a single clone has become widespread within the farm, while other potential clones have disappeared over time.

Despite more than 200 cattle having been removed from the herd after the first visit, the disease was still present within the herd, as expected in relation to the chronic nature of the disease and MAP’s environmental resistance. A part of this effect was due to the progressor animals, cows already positive to at least one of tests and further confirmed by other samplings. On the other hand, new cattle were included in the testing protocol if they were over 24 months old and had already been present in the herd, thereby being exposed to MAP. This observation confirms that the programs to eradicate the disease form infected herds require at least several years to be accomplished, leading in some cases to an eradication [19,20] or to a strong reduction in the prevalence of the disease within the herd [38].

The farm veterinarian and owner have worked closely with laboratory personnel, and adequate economic resources have been allocated to support the culling of numerous animals and the modification of facilities and management practices. These measures have enabled a progressive reduction in seroprevalence from 7.6% to 4.6% and 4.5% in the last sampling. In addition, the apparent prevalence of fecal PCR decreased (from 6.5% on the first visit to 2.8% on the second one), as did the number of low-titer or otherwise inconclusive samples, suggesting that the removal of Super and High Shedders led to an overall improvement in the paratuberculosis health situation. Emphasis is placed on the fact that seroprevalence ≤ 5 percent corresponds to the prevalence of herds with “Low risk herd” PT1 qualification, allowing the owner to consider applying for Sanitary Qualification.

Finally, the digital PCR method, used to estimate the amount of excreted MAP, proved to be a valuable tool, contributing to a more effective management and definition of the Health Management Plan of paratuberculosis in this high-prevalence infected farm. In more detail, it allowed the identification of animals with high fecal excretion, who are major contributors to environmental biocontamination. At the same time, it contributed to avoiding the slaughter of Passive Shedder animals, making the Sanitary Management Plan more effective and sustainable.

## 5. Conclusions

Overall, the results obtained indicate that the application of customized management measures particularly related to improving biosecurity and biocontainment coupled with a test-and-cull strategy, using dPCR to quickly identify animals to be removed as a matter of utmost urgency, has proven to be a successful strategy, effective in controlling infection. This can reduce the prevalence and incidence of new infections.

## Figures and Tables

**Figure 1 animals-15-02695-f001:**
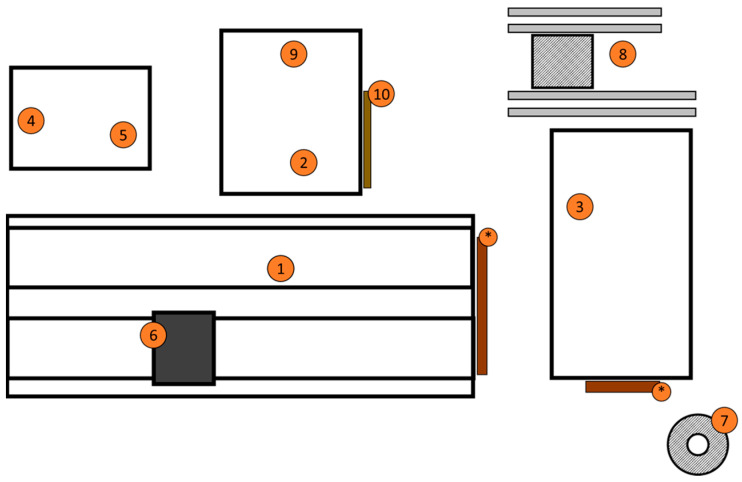
Representation of the farm layout. Point number (1) indicates the sheds dedicated to the different groups of milking cows, and number (2) shows the facilities dedicated to calves and heifers. Number (3) indicates the area dedicated to pre-parturient heifers and cows in the drying phase. The areas dedicated to the calving room and infirmary are in the same structure, but properly separated (numbers (4) and (5)). Number (6) identifies the milking area. The asterisks (*) highlight the area where the mechanical scrapers deposit the slurry, which is then transferred to the storage area (7). The feed area includes stalls for silage (8), sheds for storing straw and fodder (9), and silos for feed (10). It is important to note that both the calf’s area and the area intended for heifers were only a few dozen meters away from the site intended for adult lactating cows.

**Figure 2 animals-15-02695-f002:**
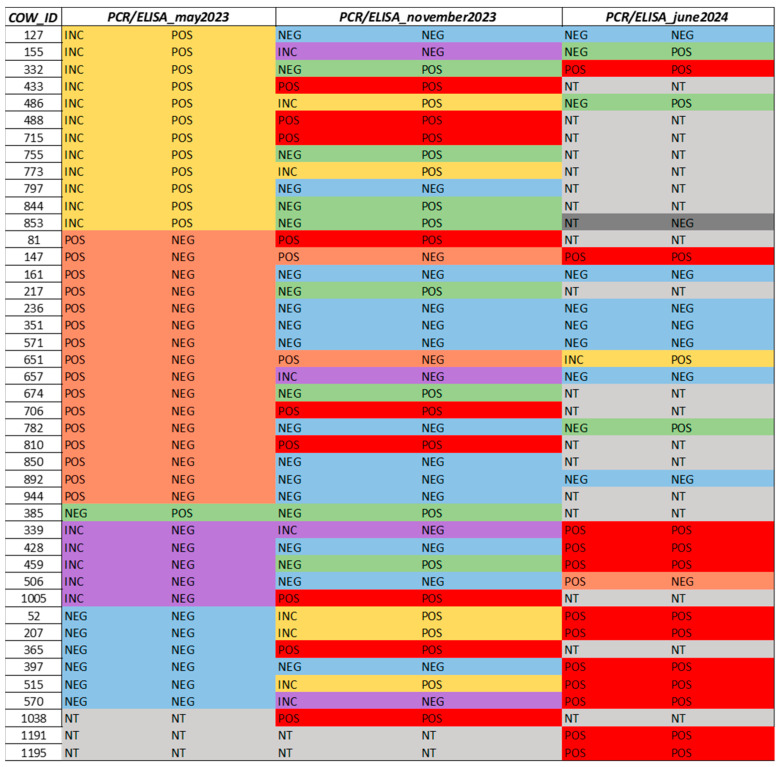
Paratuberculosis cattle results (qPCR and serology) during the study. INC: inconclusive; POS: positive; NEG: negative; NT: not tested.

**Figure 3 animals-15-02695-f003:**
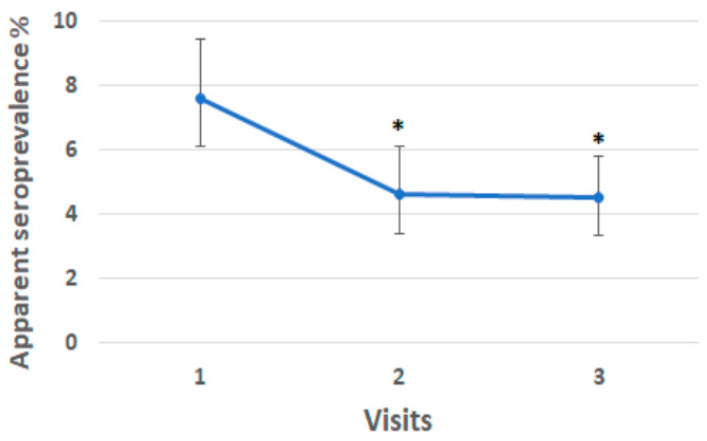
The decline in the apparent seroprevalence during the three visits. * indicates *p* < 0.05 vs. first visit.

**Figure 4 animals-15-02695-f004:**
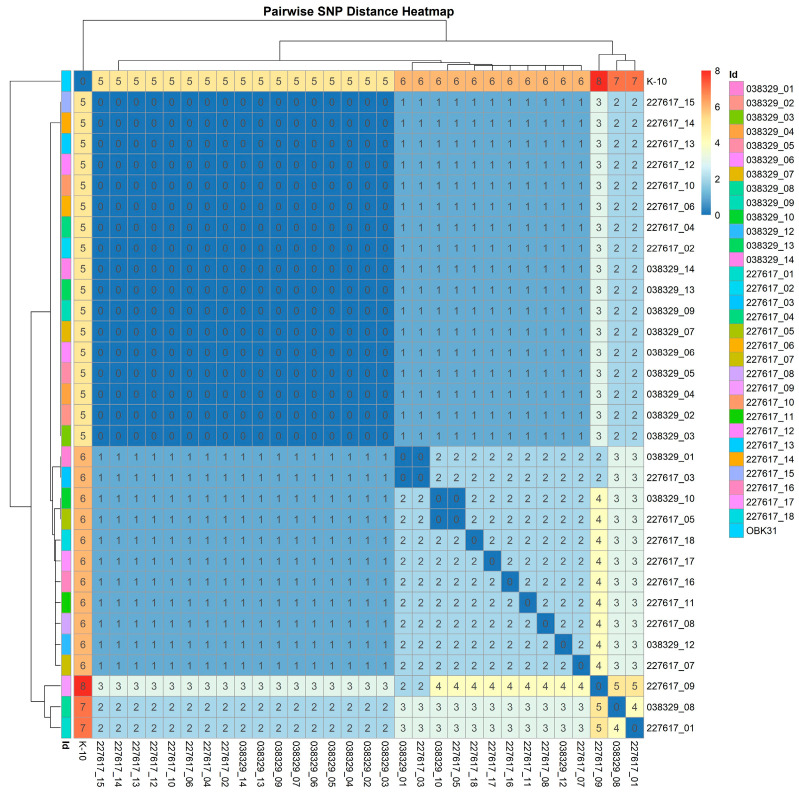
Heatmap and dendrogram showing pairwise SNP distance matrix among MAP isolates in this study.

**Figure 5 animals-15-02695-f005:**
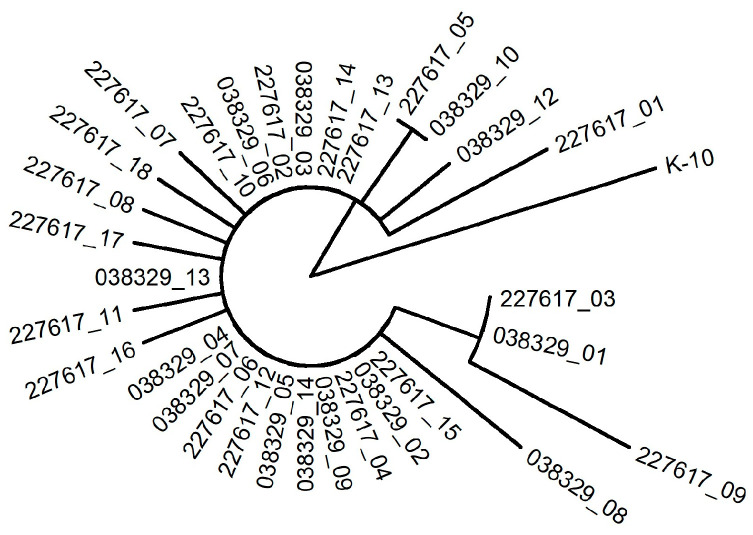
Maximum likelihood phylogenetic tree of the MAP isolates.

**Table 1 animals-15-02695-t001:** Serology and qPCR results in the first (a) and second (b) sampling. Brackets show percentages.

		qPCR								
	(a)	MAY 2023				(b)	NOVEMBER 2023			
		Pos (%)	Neg (%)	Inc ^1^ (%)	**Total**		Pos (%)	Neg (%)	Inc (%)	**Total**
**ELISA**	Pos	33 (42.3)	26 (33.3)	19 (24.4)	78		14 (31.1)	23 (51.1)	8 (17.8)	45
	Neg	33 (3.5)	753 (80.0)	155 (16.5)	941		13 (1.4)	883 (95.5)	29 (3.1)	925
	**Total**	66 (6.5)	779 (76.4)	174 (17.1)	1019		27 (2.8)	906 (93.4)	37 (3.8)	970

^1^ Inconclusive result.

**Table 2 animals-15-02695-t002:** Culture and dPCR results in the first (a), second (b), and third (c) sampling. * HS: High Shedder; ** SS: Super Shedder. CULT: cultural result; SHED_CLASS: Class of shedding.

(a)	MAY 2023								(b)	NOVEMBER 2023				(c)	JUNE 2024			
	COW_ID	CULT	SHED_CLASS	COPIES/g	COW_ID	CULT	SHED_CLASS	COPIES/g		COW_ID	CULT	SHED_CLASS	COPIES/g		COW_ID	CULT	SHED_CLASS	COPIES/g
	81	+		N.A.	674	+		1.84 × 10^4^		27	-		5.84 × 10^4^		52	+	HS	7.89 × 10^6^
	100	+		1.99 × 10^4^	677	-		2.10 × 10^4^		81	+		N.A.		147	+	HS	4.16 × 10^6^
	131	-	HS *	5.37 × 10^6^	706	+		2.09 × 10^4^		147	+		2.01 × 10^4^		207	+	HS	1.46 × 10^6^
	147	+	HS	4.18 × 10^5^	736	+		N.A.		187	+		2.09 × 10^4^		332	+		N.A.
	157	+	HS	3.05 × 10^6^	737	+	HS	1.00 × 10^5^		260	+		6.01 × 10^4^		339	+		3.75 × 10^4^
	161	-	HS	1.43 × 10^5^	740	+	HS	2.63 × 10^5^		287	+		N.A.		397	+	HS	3.70 × 10^6^
	163	-	HS	1.41 × 10^5^	741	+	SS	6.32 × 10^7^		365	+	HS	1.08 × 10^6^		428	+		N.A.
	217	+	HS	5.09 × 10^5^	747	+		3.97 × 10^4^		433	+		N.A.		459	+		3.67 × 10^4^
	236	-		5.96 × 10^4^	748	-		N.A.		453	-		2.05 × 10^4^		506	-		N.A.
	252	+		N.A.	752	+		2.13 × 10^4^		488	+	HS	2.95 × 10^6^		515	-		1.87 × 10^4^
	351	-		8.15 × 10^4^	753	-	HS	1.69 × 10^5^		603	+		N.A.		570	+		N.A.
	357	+	HS	2.44 × 10^6^	754	-		4.25 × 10^4^		651	-		6.23 × 10^4^		1191	-		5.56 × 10^4^
	471	+	HS	1.03 × 10^5^	759	+		1.87 × 10^4^		706	+	HS	2.08 × 10^5^		1195	+		8.03 × 10^4^
	473	+		N.A.	769	+	HS	5.05 × 10^5^		715	-		2.03 × 10^4^					
	485	+	SS **	7.65 × 10^7^	775	+	HS	5.84 × 10^5^		791	-		N.A.					
	487	+	HS	3.17 × 10^5^	776	+	HS	1.11 × 10^5^		810	+		N.A.					
	490	+	HS	1.80 × 10^5^	782	-		N.A.		839	+		N.A.					
	492	+		1.75 × 10^4^	783	-		N.A.		841	+	HS	1.80 × 10^5^					
	493	-		2.08 × 10^4^	800	+		4.30 × 10^4^		877	-		2.15 × 10^4^					
	494	+	HS	2.45 × 10^6^	808	+		2.15 × 10^4^		906	-		5.98 × 10^4^					
	495	+	HS	1.04 × 10^5^	810	-		N.A.		1005	+	HS	1.60 × 10^5^					
	496	-		2.08 × 10^4^	813	+	HS	6.15 × 10^5^		1038	-	HS	5.89 × 10^5^					
	499	+		2.06 × 10^4^	836	+	SS	1.17 × 10^7^		1053	+		N.A.					
	559	+		N.A.	837	+	SS	5.93 × 10^7^		1066	+		4.07 × 10^4^					
	571	-		4.00 × 10^4^	846	+	HS	4.30 × 10^5^		1074	+	HS	1.03 × 10^5^					
	591	-		N.A.	849	+		N.A.		1130	-		N.A.					
	600	+	SS	2.76 × 10^7^	850	+		N.A.		1133	+		N.A.					
	608	+		3.67 × 10^4^	851	+		5.53 × 10^4^										
	644	+		4.16 × 10^4^	892	-		2.12 × 10^4^										
	651	-		N.A.	928	+		2.08 × 10^4^										
	657	-		N.A.	944	-	HS	2.40 × 10^5^										
	668	+	HS	1.59 × 10^5^	981	-		N.A.										
	670	+		6.30 × 10^4^	987	+	SS	1.33t × 10^7^										

## Data Availability

The original contributions presented in this study are included in the article/supplementary material. Further inquiries can be directed to the corresponding author.

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
