# Peer review of "The Results After One Year of an Experimental Protocol Aimed at Reducing Paratuberculosis in an Intensive Dairy Herd"

_animals, 2025, doi:10.3390/ani15182695_

Round 1
Reviewer 1 Report
Comments and Suggestions for Authors
The comments and suggestions are included in the attached review report.

Author Response
This study describes a longitudinal investigation of paratuberculosis dynamics and control
strategies in a commercial dairy herd, combining serological surveillance with PCR testing
across multiple timepoints. The work provides practical insights into the challenges of managing
MAP in farm settings, and the field-based approach reflects the complexity of implementing
control programs under commercial conditions. However, some methodological limitations
affect the robustness and generalizability of the findings. That said, field work often involves
unavoidable logistical constraints, and the authors are encouraged to address these points
transparently to strengthen the value and interpretability of their findings. Additionally, the
manuscript would benefit from revision for language and structure to improve clarity.
A: We thank the reviewer for these positive comments and for the suggestions. We have tried our best in order to properly reply to all comments and strengthen the value and interpretability of our findings.
All changes to the manuscript are highlighted in yellow.
General comments:
Q: Title: The study period appears to span from May 2023 to June 2024 approximately one year. I
recommend adjusting this to avoid potential overstatement of the intervention duration. This
change would also provide a more accurate context for interpreting the moderate reduction in
prevalence reported.
A: We thank the reviewer for her/his advice, the title has been changed accordingly. We hope this clarifies the better the work reported in the paper.
Q: Introduction: I would discuss previous field studies trying to reduce paratuberculosis prevalence
e.g.:
Donat, K., et al. (2024). Successful control of Mycobacterium avium subspecies
paratuberculosis infection in a dairy herd within a decade A case study. Animals, 14(6), 984.
https://doi.org/10.3390/ani14060984
Gavin, W.G., Porter, C.A., Hawkins, N., Schofield, M.J., & Pollock, J.M. (2018). Johne’s disease: A successful eradication programme in a dairy goat herd. The Veterinary Record, 182(17), 483.
https://doi.org/10.1136/vr.104507
A: Thanks for the advice; the references have been added in the “Introduction” section within a brief comment about the successful eradication of paratuberculosis.
Q: Lines 147-153: A major limitation of this study is that the final sampling included only animals
positive to serology (particularly given its low sensitivity) and those that were classified as “to be cautionded”. Additionally, the manuscript does not clearly specify the criteria or rationale behind
the classification of animals as “to be cautioned”. Clarification of this category, including the definition and basis for assigning animals to it, would improve the transparency and interpretability of the study.
A: Thank you for the suggestion; due to the misleading term, we have decided to change “to be cautioned” to “to be under monitoring” and, a clearer definition, has been added to the lines 162-166. Basically, as reported also elsewhere in the manuscript, such an extensive protocol was very expensive, and as a consequence, it was possible if partially funded by other founds, not only by the owner. Taking in mind that, there were not founds enough at the end to cover the costs of all PCR analyses, so we have preferred focusing our attention on some animals. This choice has been motivated within the manuscript and it is further explained in the reply of another comment. Additionally, the concept of animals “to be under monitoring” has been further explained in the material and method section. We hope this makes the paper clearer.
Q: Line 157-158: Regarding this endpoint sampling: Finally, 1115 blood in June 2024 and after two
days, 147 fecal samples were collected.”. Given the high number of blood samples, it would be
helpful to clarify whether all samples were processed (e.g., allowed to clot, centrifuged, and
stored) and ELISAs performed within the same two-day timeframe. This appears logistically
challenging.
A: We thank the reviewer for raising this point, our laboratories routinely process at least one thousand samples per day using the ELISA method. This is one of our institution's core competencies. Anyway, we have added the information suggested (see lines 180-183). Moreover, in this specific case, samples were processed immediately upon arrival using an automated system, which increased the lab's already high throughput.
Q: Line 168-170: “monocupola” and “bicupola” are not terms used in english. Consider replacing them with more descriptive terms, such as “screening ELISA” and “confirmatory ELISA”, or specify the technical features (single-antigen vs dual-antigen formats, or single-well vs dual-well designs).
A: Thanks for the advice, the text has been changed in accordance with the reviewer's suggestion.
Q: Lines 217-221: I suggest revising the phrasing to improve clarity. It appears that the 45, 18, and
10 isolates were obtained from individual cows (different animals). However, the term “single cow” can be misleading, as it might imply that all isolates came from the same individual. Using “individual cows” or “different cows” would better convey that the isolates came from separate animals.
A: We thank the reviewer for his/her advice. We have modified the text in the hope of clarifying the sentences.
Q: Table 2: COLT à CULT? indicate the meaning of this abbreviation in the table title.
A: The abbreviation has been corrected and we have added an explanation in the table title.
Q: Section 3.5 Third sampling: For transparency and clarity, especially considering this is an important limitation of the study, it would be helpful to provide a clear justification for the reduced fecal PCR testing in this sampling.
Additionally, I could not find the results (% of positive samples) of the environmental sampling. In this sense no rationale behind the different number of environmental samples taken in each sampling is indicated.
A: Thank you to the reviewer for this comment. Regarding the decrease in the number of individual faecal samples in the third sampling, because of the lack of resources, we have been obliged of reducing the sampling extension. In more details, the owner payed only for the ELISA testing because this test is the one used in the sanitary control plan, while, we had resources just for analysing around 150 fecal samples by PCR. We therefore decided to screen all animals using serology and to collect faeces only from those that were either positive in the serology test or those that required attention despite being negative in the serology test (possible Passive Shedders).
Regarding environmental sampling, the aim was to collected MAP field isolates, so samples were taken from various areas of the farm. However, this was not done according to any probabilistic plan, but was a convenience sampling.
We have changed the text to consider also these points. In particular, in paragraph 3.5 and at the end of paragraph 2.2 explanatory sentences have been added to clarify the concept of the lack of resources and of the environmental sampling respectively.
Q: Line 365-368: revise the language, and the phrasing (“promiscuity”, animals that were negative,
to both assays?...).
A: Thanks to the reviewer for the advice, the text has been modified.
Q: Line 387-390: Revise the phrasing for clarity.
A: We thank the reviewer for this comment. Following a thorough review of the manuscript, we recognised that these sentences did not contribute significantly to the paper's meaning and we removed them.
Q: Section 3.6 Paratuberculosis apparent seroprevalence during the study: The authors report
apparent serological prevalences and confidence intervals but do not specify how these were
calculated. For clarity and reproducibility, please include a brief description of the method used
to calculate the apparent prevalence, the confidence intervals, and the statistical tests applied
to compare prevalences among visits.
Additionally, “table 3” is referred, but this table does not appear in the manuscript. Instead, it seems theat the information is shown in “figure 3”. Please review and correct the references to ensure consistency and that all cited tables and figures are included.
A: We thank the reviewer for having raised this point since we recognise adding this information improves the readability of the paper. In order to reply to this comment, a new paragraph in the material and method section, reporting all this information, has been added. Moreover, we have also further checked our data by using another software and this found a slight difference in the CI95% for the first prevalence. The figure 3 was changed to take into account the slight variation within the CI95%. Moreover, the p value for the difference in the three visits passed from 0.0080 to 0.0016 (and of course also the chi-square value associated). Also, the p value of the difference between visit 2 and 3 passed from 0.895 to 0.8657, while the rest of the manuscript remains unchanged.
Furthermore, the reference to table 3 has been removed from the text as suggested.
Q: Section 4. Discussion: several parts of paragraphs 2-5 seem more appropriate for the introduction, or they repeat information already presented there, resulting in some redundancy. I suggest revising this section to focus more directly on interpreting the study’s results.
A: We thank the reviewer for these helpful comments. According to this suggestion, we have reduced paragraphs 2–4, but we would like to keep paragraph 5 almost intact, since it is useful for introducing the use of dPCR in the context of diagnostic testing. We hope that this new version satisfies the reviewer's judgment.
Q: Supplementary Figure S1: though it is explained in the manuscript, I would add a title to the
figure and more information for example in the axis labels.
A: Thanks for the suggestion. A title has been added and the y-axis labels have been modified, according to the reviewer's advice.
Minor comments and recommended changes:
Aside from the grammatical and phrasing issues already highlighted, there are numerous minor
grammatical and language errors throughout the manuscript. I recommend thorough revision on
this aspect. A few examples:
Q: Lines 66-67: “at the moment there not available cures” à “at the moment there are no available
cures/no effective treatments are available”?
A: The text has been changed in accordance with the reviewer's suggestion.
Q: Line 69: “plan” à “plans”?
A: The text has been modified.
Q: Line 216: “Moderately shedders” à “Moderate shedders”?
A: The text has been changed in accordance with the reviewer's suggestion.
Q: Line 292-293: revise the phrasing.
A: The sentence has been revised.
Q: Line 535: “among these lasts” à “among these last”.
A: The text has been modified.
Q: Line 560: “The level of excretion of MAP from faces differs widely between excretory animals”. à “The level of MAP excretion in feces differs widely between secretory infected animals”.
A: The text has been changed in accordance with the reviewer's suggestion.
Q: Line 569: revise the phrasing.
A: The sentence has been revised.
Q: Line 611: “reformed” à “removed”?
A: The text has been changed in accordance with the reviewer's suggestion.
Reviewer 2 Report
Comments and Suggestions for Authors
The manuscript entitled "The results after two years of an experimental protocol aimed at 2 reducing paratuberculosis in an intensive dairy herd" contains useful information regarding the measures that are to be applied in a farm to control the spread of the chronic disease Johne's disease. The work carried out is exhaustive, and the authors have compiled the data in a meaningful way, but there are a few suggestions/clarifications needed:
1. The language throughout the manuscript needs to be revised, as certain areas cannot be understood properly.
2. Line numbers 584-589: Can be moved to the acknowledgement section
3. The discussion has repetition of methods, results, and already known points from the introduction. Can be rewritten about the results obtained
4. What was the status of the animals removed from the Animals removed from farm were they slaughtered or maintained in other areas?
5. What was the recommendation provided for disinfecting the environment?
6. Has the whole genome sequence data been submitted to the repository? If done, mention the accession numbers
7. What was the reference genome used for WGS data analysis and phylogenetic analysis
Author Response
The manuscript entitled "The results after two years of an experimental protocol aimed at 2 reducing paratuberculosis in an intensive dairy herd" contains useful information regarding the measures that are to be applied in a farm to control the spread of the chronic disease Johne's disease. The work carried out is exhaustive, and the authors have compiled the data in a meaningful way, but there are a few suggestions/clarifications needed:
A: We thank the reviewer for these positive comments and for the suggestions. We have tried our best in order to properly reply to all comments and strengthen the value and interpretability of our findings.
All changes to the manuscript are highlighted in yellow.
Q: The language throughout the manuscript needs to be revised, as certain areas cannot be understood properly.
A: We thank the reviewer for this helpful comment; the language has been revised with the help of a native English speaker, in the hope of making the manuscript more comprehensible.
Q: Line numbers 584-589: Can be moved to the acknowledgement section
A: The sentence has been modified (see lines 575-579); however, we have decided to keep it in the manuscript to emphasise the importance of the approach used. We hope that this new version satisfies the reviewer's judgment.
Q: The discussion has repetition of methods, results, and already known points from the introduction. Can be rewritten about the results obtained
A: We thank the reviewer for these helpful comments. In agreement with your advice and another comment from reviewer 1, we have reduced and modified the “discussion” section in an effort to make it more understandable and clearer.
Q: What was the status of the animals removed from the Animals removed from farm were they slaughtered or maintained in other areas?
A: Thank you for your question. We recommend sending animals intended for slaughter to the abattoir, and not selling them for any other purpose; in line 360 we have added the information requested in order to make the sentence clearer.
Q: What was the recommendation provided for disinfecting the environment?
A: We are a little bit confused about this question because we have carefully checked the manuscript but did not find any advice relative to specific use of disinfectants. Have we overlooked any important details? We would be happy to discuss this matter further, but we did not discuss the use of specific disinfectants that have proven activity against MAP with the farmer.
Well, as reported in lines 464-466, we generically suggested to improve hygene in the farm. this farmer and his veterinary didn’t ask any suggestions; usually when asked for advice on the type of disinfectants to use against MAP in the farm, we recommend orthophenylphenate and provide bibliography with informations on disinfectants considered effective (e.g.: J.R. Stabel, A. Turner, M. Walker, 2020; “An eco-friendly decontaminant to kill Mycobacterium avium subsp.Paratuberculosis” Journal of Microbiological Methods 176 (2020) 106001.)
Q: Has the whole genome sequence data been submitted to the repository? If done, mention the accession numbers
A: The whole genome sequence data has been submitted to the repository; the accession numbers has been added in the “materials and methods” section (see line 263)
Q: What was the reference genome used for WGS data analysis and phylogenetic analysis
A: The reference genome used for WGS data analysis and phylogenetic analysis is the MAP genome K10. This information was already reported in the manuscript in “materials and methods” section and in the “results” section.
Round 2
Reviewer 2 Report
Comments and Suggestions for Authors
The authors have carried out the corrections